# Effect of Home-Based Tele-Pilates Intervention on Pregnant Women: A Pilot Study

**DOI:** 10.3390/healthcare10010125

**Published:** 2022-01-08

**Authors:** Ah-Hyun Hyun, Joon-Yong Cho, Jung-Hoon Koo

**Affiliations:** Department of Exercise Biochemistry and Exercise, Korea National Sport University, Seoul 05541, Korea; knupe838@knsu.ac.kr (A.-H.H.); chojy86@knsu.ac.kr (J.-Y.C.)

**Keywords:** COVID-19, pregnancy, physical exercise, hip flexion, hip abduction, pelvic tilt

## Abstract

Pilates is effective for training the core muscles and stabilizing the hip joints, which provides relief from pelvic pain and low back pain during pregnancy. However, there are no specific guidelines on appropriate physical exercises for pregnant women due to the current pandemic. We aimed to apply the exercise standard proposed by the American College of Obstetricians and Gynecologists to home-based tele-Pilates exercise (HTPE), to determine its effect on the physical and mental health of pregnant women. We randomly divided the subjects into the following two groups who completed 8 weeks of HTPE (50 min/day, 2 days/week): (a) Pilates exercise (PE, *n* = 7) and (B) non-Pilates exercise (CON, *n* = 7). HTPE was performed by adjusting the program every 3 weeks, based on pain and physical fitness levels. We measured body composition, muscles of the hip joint, pelvic tilt, Oswestry Disability Index (ODI), and Pittsburgh Sleep Quality Index (PSQI), before and after HTPE. Following HTPE, while the percentage of body fat and body mass index had significantly decreased, the body fat mass did not change in the PE group (*p* < 0.05). The PE group showed an increase in strength of the left and right hip flexion and hip abduction, compared to the CON group (*p* < 0.01). The ODI and PSQI were significantly decreased in the PE group (*p* < 0.05). Therefore, the 8-week HTPE program is an effective exercise for pregnant woman that reduces body fat metabolism and strengthens muscles of the hip joint, thus alleviating pregnancy-induced low back pain and insomnia.

## 1. Introduction

Coronavirus disease 2019 (COVID-19), a viral respiratory disease, has caused social, economic, and cultural disruption across the world. COVID-19 is highly contagious and causes fatal damage to elderly people with underlying conditions [1]. Pregnant women are classified as another high-risk group and are predicted to be exposed to great danger during the COVID-19 pandemic [2,3]. This can be attributed to the high incidences of miscarriage, premature birth, immature fetal growth, kidney failure, and vascular disease during the 2003 SARS outbreak. Although some studies have reported on the susceptibility of pregnant women to pneumonia and respiratory infections during the current pandemic [4], there are limited studies on COVID-19 and pregnant women. Thus, it is difficult to establish obstetric prevention guidelines.

During pregnancy, women undergo major changes in their body mechanics, physiology, and psychology. It is characterized by severe anterior pelvic tilt and lumbar curve changes due to fetal growth and an increased abdominal size [5,6]. The later stages are characterized by tension in the hip flexors and lumbar muscles, and weakening of the abdominal and hip joint muscles. These eventually lead to pelvic pain, thus resulting in body instability and low back pain [7,8]. In addition to the stress caused by the aforementioned physical changes, the psychological anxiety among pregnant women regarding the COVID-19 pandemic can further increase their stress. Increased anxiety and stress in potential mothers can cause side effects such as nausea, vomiting, loss of weight, and depression. These, in turn, can induce diabetes, hypertension, and obesity [9]. A previous study has reported a two-fold increase in anxiety disorders in pregnant women after COVID-19. This has been attributed to stress caused by hospital visits for obstetric exanimations, fetal infections during childbirth, and social isolation due to quarantine [10]. Furthermore, a study analyzed the Pittsburgh Sleep Quality Index (PSQI) in 166 pregnant women during the COVID-19 pandemic and reported that 88% of the pregnant women experienced difficulty in sleeping [10]. This necessitates a therapeutic approach to alleviate their high stress and anxiety about COVID-19.

The American College of Obstetricians and Gynecologists (ACOG) recommends non-physical therapies, such as meditation, breathing, music, and reading, and physical activities, such as yoga, swimming, and walking for at least 150 min weekly to maintain the health of the fetus [11]. Physical activity during pregnancy effectively enhances the physical fitness of the mother and relieves pain. Moreover, it facilitates short contractions during childbirth, rapid postpartum recovery, and diet [12]. However, the participation of pregnant women in physical activity has sharply declined due to pandemic-mediated environmental limitations. Nevertheless, considering the impact of physical exercise on improving anxiety and sleep disorders in pregnant women, an alternative program to increase their daily physical activity in the current environment should be initiated [13]. Pregnant women who performed voluntary exercise after COVID-19 were shown to have a lower index of depressive disorder than the control group with a sedentary lifestyle. Moreover, the insomnia severity index was alleviated in the group that performed yoga [14]. Therefore, physical activity can partially alleviate the psychological instability in pregnant women in the context of COVID-19. However, the mechanism underlying psychological stability is unclear. Moreover, there is insufficient research on the mechanism of relieving low back pain in pregnant women (muscles of the hip joint and pelvic tilt) and related pain indicators. The aforementioned women performed voluntary exercise independently during COVID-19. Hence, there is a lack of motivation for exercise and corrective feedback on the exercise method and programs, compared to face-to-face exercise. This necessitates the need for an exercise program that not only improves the muscles of the hip joint in pregnant women but also provides detailed feedback on exercise in the context of COVID-19.

Pilates not only improves basic physical strength through a combination of aerobic and anaerobic exercise, but also stabilizes the hip joint by strengthening the deep muscles close to the spine. This, in turn, alleviates physical discomfort, such as low back pain and pelvic pain [15]. Previous studies have reported on increased physical strength and hip joint muscle strength in pregnant women who participated in Pilates, and a significant decrease in visual analog scale scores (a measure of back pain) [16,17]. However, it has been difficult for pregnant women to participate in exercise due to the closure of sports facilities and the recommendations to stay at home during the COVID-19 pandemic. This calls for a program that can solve environmental constraints while maintaining the beneficial effects of Pilates.

Following COVID-19, non-face-to-face remote systems have been widely used in industrial, educational, and cultural aspects through the web service of IT platforms. The culture of non-contact consumption has dramatically increased the home-training population, where people participate in online workouts at home. However, the abuse of unverified information and content can impair exercise-related judgments. Participation in physical activity without proper knowledge of the precautions can lead to serious injuries [18]. Pain and severity in pregnant women vary according to the number of weeks of pregnancy. This necessitates a program that considers the posture to protect weakened joints, breathing for childbirth, prevention of falls, contraindications, and a two-way communication method that provides real-time education. Non-face-to-face remote exercise is generally limited to programs used to prevent muscle loss in the elderly [19,20], for rehabilitation in sports medicine [21,22], and in patients with other diseases [23,24]. Therefore, the participation of physically inactive pregnant women in home-based tele-pilates exercise (HTPE) will enable them to overcome COVID-19-associated environmental constraints and control pelvic floor muscle depression and pain. We aimed to verify whether 8 weeks of HTPE partially improves the pelvic tilt and muscles of the hip joint, low back pain, and insomnia during pregnancy in the COVID-19 context. The identification of HTPE-associated positive factors will present a new paradigm of exercise suitable for pregnant women who voluntarily practice social distancing due to limitations in the activity radius and changes in body shape in the post-COVID-19 era.

## 2. Materials

### 2.1. Subject

We selected the subjects from among the women registered at the ‘C’ Cultural Center in Bundang, Gyeonggi-do, Korea, who completely understood the purpose of the study and offered their voluntary participation and consent. The selected subjects included pregnant women under the age of 45 years, at 20–24 weeks of single-fetus pregnancy. Moreover, they did not receive medications and did not participate in home-training. This study was a cluster randomized controlled experiment. We used a lottery method for randomization, and all subjects chose a mini ball marked with 1 or 2. A total of 18 subjects were classified into the control group (CON, *n* = 9) and the exercise group (PE, *n* = 9). We excluded four subjects (*n* = 2 per group) who expressed discomfort while participating in the exercise and dropped out (CON (*n* = 7), 34.14 ± 3.82 years, 63.57 ± 3.57 kg, 37.72 ± 3.82% body fat; PE (*n* = 7), 31.71 ± 3.03 years, 67.44 ± 5.18 kg, 38.85 ± 5.53% body fat). A comparison of the reference values between the groups was performed by obtaining Cohen’s r values. Following approval from the Korean National Sports University (1263-202103-HR-002-01), we requested the subjects to fill out a written consent form and all procedures performed herein conformed to the principles outlined in the Declaration of Helsinki.

### 2.2. HTPE Program

The HTPE program consisted of warm-up, main, and cool-down exercises, and was conducted for 50 min/day, twice a week, for a total of 8 weeks according to the ACOG (Figure 1). A 20 s break was allotted between the exercises. Moreover, the exercise intensity was maintained at a 10–13 rating of perceived exertion (RPE) by the ACOG (50–60% of the maximum heart rate) and Borg’s scale. We gradually increased the exercise intensity every 3 weeks, based on the subject’s physical fitness level and pain status. Table 1 summarizes the HTPE program. We restricted the number of participants to <10 to help the instructor observe the movement and pain level of all subjects by reflecting on the characteristics of non-face-to-face exercise. Their incorrect postures were corrected and the RPE was monitored to maintain an appropriate intensity. Furthermore, the subjects were able to maximize the mutual exercise effect by adjusting the distance from the instructor by using a television or a tablet in their respective spaces, receiving feedback on their posture, and real-time conversations on the degree of discomfort, before and after the movement.

### 2.3. Body Composition

The subjects maintained a fasting state and removed metal accessories before their height and body measurement using a DS-103M automatic height meter (Jenix Co., Seoul, Korea) and a InBody H20B body composition analyzer (Biospace Co., Seoul, Korea), respectively. Both hands and feet were disinfected before the measurement. They were directed to stand on the platform and hold the handles with the electrodes in each hand. During the test, both arms were kept slightly open to prevent contact with the torso. Moreover, talking was prohibited and they were requested to maintain comfortable breathing. The measurement factors included body weight (BW, kg), body fat mass (BFM, kg), skeletal muscle mass (SMM, kg), body mass index (BMI, kg/m^2^), percentage of body fat (PBF, %), and fat free mass (FFM, kg).

### 2.4. Pelvic Tilt

A posturemeter (Spomedic healthcare, Seoul, Korea) was used to measure the pelvic tilt, such as the coronal plane tilt (CPT) and sagittal plane tilt (SPT) [25]. After standing upright, the subjects lowered their arms comfortably and stared straight ahead. The examiner marked the anterior superior iliac spine in the front and posterior superior iliac spine at the back of the pelvis with stickers. The posturemeter was placed on the marked anterior iliac spine to measure the CPT tilt. In contrast, the SPT was measured by applying a posturemeter on the marked poles on the anterior and posterior iliac spines in the sagittal plane. The posturemeter protractor was set to 0 degrees before the measurement, and the average of two measurements was noted and recorded.

### 2.5. Muscles of the Hip Joint

Muscles of the hip joint that are involved in right hip flexion (RHF), right hip abduction (RHA), left hip flexion (LHF), and left hip abduction (LHA) were measured using a manual muscle strength meter (HOGGAN PROOF Preferred, HOGGAN HEALTH, Salt Lake City, UT, USA). An isometric muscle test was used for the measurement and conducted using the active straight leg-raising method with the subject in a supine position [26]. The subjects practiced once, and the measurement was performed twice. The examiner measured the maximum muscle strength range at which the subjects did not feel pain with their trunk fixed. The muscle strength was measured by fixing a manual muscle strength meter on the right ankle and requesting the subjects to raise their foot to the ceiling to the maximum extent possible. This enabled the measurement of the RHF and LHF. For measuring the RHA and LHA, the examiner fixed the manual muscle strength tester to the upper ankle while the subjects were laid on their side with the head comfortably placed on one arm, and the knees bent to fold in both feet. They were then asked to lift the upper foot towards the ceiling on the side of the hip joint to measure the muscle strength. All measurements were repeated twice, and the average value was used for the analysis. A 30 s break was allotted for each muscle group. The unit of the instrument was 1 lb with a ±1% margin of error.

### 2.6. ODI Test

The Oswestry Disability Index (ODI) test is a questionnaire developed to measure the level of symptoms in patients with lower back pain. Furthermore, it enables the verification of pain-associated degree of dysfunction. The evaluation items consist of 10 questions on pain management, personal management, walking, standing, sitting, sleep, and social life, and each question is evaluated on a scale of 0–5. The scores for each item are added to obtain the total score, which is then divided by 50 and multiplied by 100 to obtain a percentage. Values in the range of 0–20%, 21–40%, 41–60%, 61–80%, and 81–100% indicate minimal disability, moderate disability, severe disability, paralyzed, and patients who must be bed-bound, respectively [27]. The test was scored twice before and after the experiment, and the total score was recorded.

### 2.7. PSQI Test

We used the PSQI developed and modified to evaluate sleep quality [28]. PSQI is a self-reported questionnaire that measures sleep quality and sleep disturbances over the past month. It includes 18 items and seven primary factors, namely sleep quality, sleep delay, sleep duration, habitual sleep efficiency, sleep disability, sleep medication use, and daytime dysfunction. The scale is evaluated based on a total score, and each item is scored from 0 to 3. The total score is obtained by summing the scores of the aforementioned seven factors. The total score ranges from 0 to 21. Higher the total score, the lower the sleep quality is. A total score ≤ 5 indicates sound sleep. In contrast, a score ≥8 indicates a sleep problem. While the reliability of the PSQI tool has been measured to obtain a Cronbach’s alpha of 0.85, we obtained a Cronbach’s alpha of 0.81 [29].

### 2.8. Statistical Analyses

We analyzed the differences in the body composition, pelvic tilt and muscles of the hip joint, ODI, and PSQI before and after HTPE using SPSS 22.0. The result of the normality test revealed unsatisfactory normal distribution. Therefore, statistical analyses were conducted using a non-parametric test method. The difference between the groups was analyzed using the Mann–Whitney U test, based on the average difference obtained through the change-score analysis. The difference between the pre- and post-test status within the groups was analyzed using the Wilcoxon signed-rank test. All statistical values are presented as median (interquartile range), and the level of significance for was set to *p* < 0.05.

## 3. Results

### 3.1. Effect of HTPE on the Body Composition

There was no statistically significant difference in the pre- and post-body composition between the groups (Table 2). All body composition factors of the CON group increased significantly post-HTPE (BW: *p* = 0.018, SMM: *p* = 0.028, TFM: *p* = 0.018, PBF: *p* = 0.018, BMI: *p* = 0.018, and FFM: *p* = 0.018). While PBF and BMI had significantly decreased (PBF: *p* = 0.043 and BMI: *p* = 0.018), other factors had significantly increased (BW: *p* = 0.018, SMM: *p* = 0.018, and FFM: *p* = 0.018); no significant difference was observed in the BFM in the PE group.

### 3.2. Effect of HTPE on Muscles of the Hip Joint

The muscle strength in the hip joint, and the RHF, RHA, LHF, and LHA of all muscles were significantly increased in the PE group, compared to the CON group (RHF: *p* = 0.001, RHA: *p* = 0.001, LHF: *p* = 0.001, and LHA: *p* = 0.001, Table 3). Moreover, all muscles in the PE group showed a significant increase after HTPE (RHF: *p* = 0.180, RHA: *p* = 0.180, LHF: *p* = 0.180, and LHA: *p* = 0.180). Although a decreasing trend in all muscles was observed in the CON group, the difference was not significant.

### 3.3. Effect of HTPE on the Pelvic Tilt

There was no substantial difference in the amount of change in the pelvic tilt such as CPT and SPT between the groups (Table 4). Moreover, there was no statistically significant difference in the CPT in the PE group, though we observed an increase in the SPT (SPT: *p* = 0.017). However, the CON group revealed increased inclination in both the CPT and SPT (CPT: *p* = 0.038 and SPT: *p* = 0.017).

### 3.4. Effect of HTPE on the ODI and PSQI

Changes in the ODI and PSQI were significantly decreased in the PE group, compared to the CON group (ODI: *p* = 0.001 and PSQI: *p* = 0.001, Figure 2). Following HTPE, we observed a significant decrease in the ODI and PSQI in the PE group (ODI: *p* = 0.028 and PSQI: *p* = 0.161). There was no significant difference in the ODI in the CON group; however, the PSQI was increased (*p* = 0.017).

## 4. Discussion

Researchers have recently emphasized the importance of exercise for pregnant women. This can be attributed to the impact of regular exercise on reducing anxiety and stress during pregnancy. Moreover, it exerts anti-inflammatory and antiviral effects [30]. Various methods have been proposed to overcome the limitations of physical activity during pregnancy in the midst of the COVID-19 pandemic [31]. Non-face-to-face online alternative programs are used in some areas of rehabilitation. Nonetheless, there is little evidence regarding their effectiveness in pregnant women. Therefore, we determined the positive effect of HTPE on various physiological changes, such as the body composition, pelvic tilt, and muscles of the hip joint during pregnancy, and the resulting low back pain and insomnia.

Following HTPE, there was no statistically significant difference in body composition changes among pregnant women. This phenomenon is commonly caused by increased insulin and lipid hormones during pregnancy. Our results were similar to those of previous studies that reported weight gain in pregnant women [32]. Weight gain during pregnancy is inevitable. However, excessive weight gain causes complications such as hypertension, diabetes, and hyperlipidemia, thus necessitating proper diet control and exercise [33,34]. Pilates is reportedly effective at controlling body weight and body fat in pregnant women and facilitates childbirth by increasing the skeletal muscle mass [35]. Thus, we aimed to confirm whether HTPE triggered changes in body fat metabolism. There was no significant difference in the BFM in the PE group. The PBF and BMI had decreased after HTPE, thus indicating that HTPE suppressed the increase in body fat mass during pregnancy. This may be attributed to the ACOG-based intensity of HTPE that is sufficient to reduce the fat mass. Our results were similar to those of previous studies that have reported loss in body fat after Pilates exercise. In addition, resistance movements, such as the contraction and relaxation of skeletal muscles, along with a Pilates-induced reduction in body fat increase the SMM and muscle strength [35]. Accordingly, an HTPE-mediated increase in the SMM would likely occur. However, SMM was significantly increased in both groups post-HTPE. This might be associated with a partial increase in skeletal muscle, a factor of body composition, due to increased body weight during pregnancy. Thus, we failed to interpret our results as an HTPE-mediated increase in SMM. However, we observed a significant increase in the muscle strength of the hip joint in the PE group, compared to the CON group. This was consistent with the findings in previous studies that mentioned that Pilates exercise strengthens the pelvic muscles with core exercise [36]. Therefore, HTPE is considered an effective form of exercise that can reduce the elevated body fat metabolism during pregnancy and strengthen the muscles of the hip joint.

The anterior tilt of the hip joint increases during the later stages of pregnancy, thus, collapsing the pelvic inclination, and causing back pain with spinal deformity [37,38]. The CPT and SPT significantly increased in the CON group, which was consistent with the findings of previous studies that have reported on pelvic tilt deformity in pregnant women [39,40]. However, there was no significant difference in the CPT in the PE group. Hence, HTPE possibly alleviated the pelvic tilt during pregnancy. Our findings were consistent with those of previous studies that reported on pelvic tilt relief through Pilates exercise [41]. In contrast, changes in anterior and posterior pelvic tilt in the SPT increased with the gestation period in both groups. This was consistent with the finding of a study that mentioned that hip flexors become shortened during pregnancy and the lumbar curve becomes severe, which results in falls due to body imbalance. In addition, loss of balance is usually observed in late pregnancy [42]. In our study, although the difference was not statistically significant, the amount of change in the SPT tended to decrease in the PE group, compared to the CON group. Therefore, HTPE is considered an effective exercise method that can partially alleviate pelvic imbalance during pregnancy. However, the mechanism underlying the control of pelvic imbalance is unclear. This may be related to the strengthening of hip joint muscles by HTPE. Interestingly, we confirmed an increase in the muscle strength of the hip joint following HTPE. Thus, HTPE supposedly improves pelvic tilt imbalance by strengthening the muscles around the pelvis during pregnancy.

Most pregnant women suffer from insomnia during pregnancy due to back pain. The post-ODI and PSQI scores increased in the CON group compared with before. An increase in stress can have a negative effect on the mother and child. Thus, an appropriate treatment is required to relieve low back pain. Considering the impact of HTPE on increasing pelvic strength and improving the pelvic tilt, our study findings may facilitate relieving low back pain and related insomnia during pregnancy. The ODI and PSQI scores significantly decreased in the PE group post-HTPE. Therefore, Pilates strengthens the core muscles, thus stabilizing the radius and alleviating low back pain and insomnia during pregnancy [43,44]. In other words, HTPE alleviates low back pain and insomnia during pregnancy. This can be attributed to the strengthening of the hip joint muscles and relieving the pelvic tilt imbalance. Therefore, HTPE is an effective exercise method that can partially alleviate low back pain and sleep disturbances that occur during pregnancy. The effects of exercise on pregnant women in this study point toward the benefits of an online learning platform for sports [45]. This is attributable to the development of applications that facilitate real-time interactions. In the future, subjects should be able to use epidemiological and physiological data to practice through online training using artificial intelligence such as Metabus.

The limitations of our study include our inability to mobilize larger subjects due to its design that required pregnant women. Moreover, some mothers were unable to participate in the exercise due to personal circumstances that arose during pregnancy. In addition, we could not recruit a large number of participants to provide individual communication about the exercise and accurate feedback through video. This can be attributed to the non-face-to-face exercise program. Consequently, we could not generalize the results. This would enable the management of larger subjects in the future. Hence, it is necessary to use remote technology to develop an exercise program specialized for pregnant women. In addition, it is considered necessary to further study whether this non-face-to-face exercise program can be applied to postpartum women who may be vulnerable to depression as well as during pregnancy.

## 5. Conclusions

Eight weeks of HTPE resulted in a limited increase in body fat and increased the pelvic strength, thereby reducing the pelvic tilt caused by pregnancy. In addition, HTPE reduced back pain and relieved sleep disturbances resulting from pregnancy. Therefore, HTPE may be considered an effective exercise for enhancing the physical and mental changes that occur in pregnant women, similar to face-to-face Pilates exercise. Furthermore, HTPE will likely serve as a suitable remote technology exercise program that can alleviate the anxiety and physical discomfort experienced by pregnant women in future pandemics.

## Figures and Tables

**Figure 1 healthcare-10-00125-f001:**
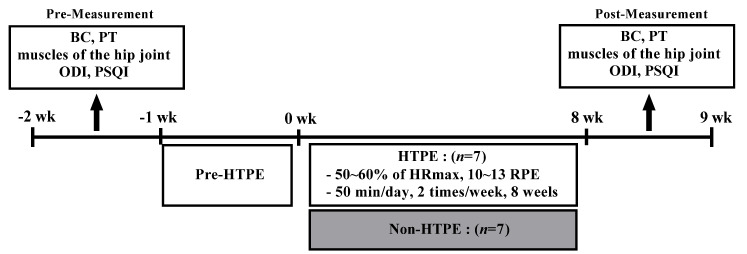
Experimental design. BC: body composition, PT: pelvic tilt, ODI: Oswestry Disability Index, PSQI: Pittsburgh Sleep Quality Index, HTPE: home-based tele-pilates exercise, HRmax: maximal heart rate, RPE: rating of perceived exertion.

**Figure 2 healthcare-10-00125-f002:**
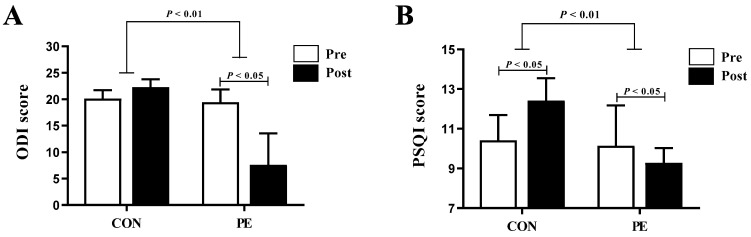
HTPE reduced ODI and PSQI score in Korean pregnant woman. (**A**) ODI score and (**B**) PSQI score (*n* = 7 per group). Error bars represent mean ± SD. CON: non-pilates exercise, PE: Pilates exercise, ODI: Oswestry Disability Index, PSQI: Pittsburgh Sleep Quality Index.

**Table 1 healthcare-10-00125-t001:** HTPE program.

Modes	Contents	Time (min)	Reps, Set, and Rest	RPE
Warm-up	Low-impact stretching and Breathing	10		10
Main exercise	Level 1: 1–3 week Arm circle, Cat cow, Bridge, Clam, Half-spine rotation, Leg circles, Half-squat Level 2: 4–6 week Donkey kick, Hip hinge, Leg side up, Half-saw, Half-lunges, Kneeing push-up Level 3: 7–8 week Half-lunge twist, Side lateral raise, Squat, Low-impact down dog, Deep breathing	30	12–15 reps × 3 set 10 s rest between sets	11–13
Cool-down	Total body stretching	10		10

**Table 2 healthcare-10-00125-t002:** Body composition.

	CON (*n* = 7)		PE (*n* = 7)		Diff (Post-Pre)	
	Pre	Post	Pre	Post	*p*	Cohen’s d
BW (kg)	64.50 (8.70)	69.90 (8.90) *	69.10 (13.00)	72.50 (16.20) *	0.456	0.337
SMM (kg)	27.70 (9.00)	28.20 (10.70) *	23.40 (8.70)	29.50 (8.10) *	0.805	0.228
BFM (kg)	23.70 (12.10)	24.90 (10.90) *	25.10 (8.00)	29.50 (8.10)	0.259	0.706
PBF (%)^)^	39.30 (14.20)	40.40 (9.80) *	37.90 (11.10)	37.70 (14.60) *	0.209	0.768
BMI (kg/m^2^)	23.50 (1.60)	25.50 (2.10) *	25.30 (5.10)	26.90 (4.80) *	0.383	0.706
FFM (kg)	39.70 (10.40)	39.90 (8.80) *	40.60 (13.60)	44.50 (18.20) *	0.128	0.861

Values are presented as median (interquartile range) (*n* = 7 per group). Main time effect: * *p* < 0.05, pre- versus post-HTPE period in the within groups. CON: non-pilates exercise, PE: pilates exercise, BW: body weight, SMM: skeletal muscle mass, BFM: body fat mass, PBF: percentage of body fat, BMI: body mass index, FFM: fat free mass.

**Table 3 healthcare-10-00125-t003:** Muscle strength in the hip joint.

	CON (*n* = 7)		PE (*n* = 7)		Difference (Post-Pre)	
	Pre	Post	Pre	Post	*p*	Cohen’s d
RHF (lbs) ##	3.80 (2.40)	3.60 (2.30)	4.30 (2.20)	6.40 (3.80) *	0.001	2.939
RHA (lbs) ##	3.50 (0.80)	3.20 (0.70)	5.00 (3.60)	7.10 (1.70) *	0.001	6.958
LHF (lbs) ##	3.60 (1.20)	3.40 (1.00)	4.80 (3.00)	6.80 (2.90) *	0.001	4.482
LHA (lbs) ##	3.30 (0.80)	3.20 (1.40)	4.30 (1.50)	7.00 (1.80) *	0.001	6.889

Values are presented as median (interquartile range) (*n* = 7 per group). ## *p* < 0.01 change (post–pre) between groups. Main time effect: * *p* < 0.05, pre- versus post-HTPE period within groups. CON: non-pilates exercise, PE: pilates exercise. RHF: right hip flexion, RHA: right hip abduction, LHF: left hip flexion, LHA: left hip abduction.

**Table 4 healthcare-10-00125-t004:** Pelvic tilt.

	CON (*n* = 7)		PE (*n* = 7)		Diff (Post-Pre)	
	Pre	Post	Pre	Post	*p*	Cohen’s d
CPT (°)	2.00 (0.00)	3.00 (2.00) *	2.00 (1.00)	2.00 (2.00)	0.259	0.681
SPT (°)	22.00 (8.00)	28.00 (5.00) *	24.00 (18.00)	28.00 (12.00) *	0.456	0.774

Values are presented as median (interquartile range) (*n* = 7 per group). Main time effect: * *p* < 0.05, pre- versus post-HTPE period in the within groups. CON: non-pilates exercise, PE: Pilates exercise, CPT: coronal plane tilt, SPT: sagittal plane tilt.

## Data Availability

The datasets used and analyzed during the current study are available from the corresponding author on reasonable request.

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
