# Peer review of "Effect of Home-Based Tele-Pilates Intervention on Pregnant Women: A Pilot Study"

_healthcare, 2022, doi:10.3390/healthcare10010125_

Round 1
Reviewer 1 Report
dear Authors, thank you for being amenable to all my suggestions. I think that the paper has clearly improved in quality and clearness. I appreciated the hint to a potential application (of the proposed approach, irrespectively from the exercise typology) for distance teaching and coaching of sports. This point assumes even a greater importance during this pandemic period. But I strongly recommend the authors to add some references for giving a scientific base to that statement (page 9, "The effects of exercise on pregnant women in this study point toward the benefits of an online learning platform for sports.". I would suggest, for instance: https://doi.org/10.1007/978-3-030-31284-8_17 and http://doi:10.4018/978-1-5225-5912-2.ch002
Author Response
First of all, we sincerely thanked you for your insightful suggestion. We have added the reference you recommended. (see page 9 lines 7, ref-49).
[49] Picerno P., Pecori R., Raviolo P., Ducange P. (2019) Smartphones and Exergame Controllers as BYOD Solutions for the e-tivities of an Online Sport and Exercise Sciences University Program. In: Burgos D. et al. (eds) Higher Education Learning Methodologies and Technologies Online. HELMeTO 2019. Communications in Computer and Information Science, vol 1091. Springer, Cham. https://doi.org/10.1007/978-3-030-31284-8_17

Reviewer 2 Report
I do not see a cover letter to this paper, which would perhaps clarify its story. Reviews and responses to them are included, which may suggest that the text was considered in another journal. Corrections as suggested by the 3 reviews have been made and it can be seen that this improves the quality of the publication. My additional comments are minor.
Still the main problem is the small size of both groups which affects the statistical analyses. Perhaps it would be worthwhile to look for tests for small samples and not only non-parametric tests.
Table 5 has a very low value, so it could be included in the appendix or omitted altogether. N is not given in the title, so perhaps these are two groups combined.
PSQI test is misleading, as usually high quality means better outcome. Please remind in results section than high PSQI means worse sleep quality. And thus the imropvement in PE group is visible on the Figure 2.
It would also be better to annotate the p<0.001 value in Figure 2 instead of the ## marking, which is not readable at first glance.
Since this is a preliminary study, it is worth adding further directions. Are there plans to implement such a program? What changes are worth making? In a study with a larger sample, a mixed model could also be proposed. In this study there was only pre-post comparison in each group separately.
Author Response
1. I do not see a cover letter to this paper, which would perhaps clarify its story. Reviews and responses to them are included, which may suggest that the text was considered in another journal. Corrections as suggested by the 3 reviews have been made and it can be seen that this improves the quality of the publication. My additional comments are minor.
Response: We sincerely thanked you for your valuable time to review our manuscript. First of all, we are really apologize for attaching another journal’s cover letter. As you mentioned, we have attached the cover letter again. Just in case, we have also described it down here.
Dear Dr. Rahman Shiri
I wish to submit an original article for publication in healthcare titled “Effect of home-based tele-Pilates intervention on pregnant women: a pilot study.” The paper was co-authored by Ah-Hyun, Hyun and Joon-Yong, Cho. This study aimed to apply the exercise standard proposed by the American College of Obstetricians and Gynecologists to the home-based tele-Pilates exercise (HTPE), to determine their effect on the physical and mental health of pregnant women. We randomly divided pregnant women into the following two groups that completed 8 weeks of HTPE (50 min/day, 2 days/week): (a) Pilates exercise group (PE, n=7) and (B) non-Pilates exercise (CON, n=7). We measured the body composition, muscles of the hip joint, pelvic tilt, Oswestry Disability Index, and Pittsburgh Sleep Quality Index, before and after HTPE. We believe that our study makes a significant contribution to the literature because HTPE effectively reduced the body fat metabolism and strengthened muscles of the hip joint, thus alleviating pregnancy-induced low back pain and insomnia. Further, we believe that this paper will be of interest to the readership of your journal because the recognition of HTPE-associated positive factors will present a novel paradigm of exercise, suitable for pregnant women who voluntarily practice social distancing because of limitations in the activity radius and changes in body shape in the post-COVID-19 era.
This manuscript has not been published or presented elsewhere in part or in entirety and is not under consideration by another journal. All study participants provided informed consent, and the study design was approved by the Korean National Sports University. We have read and understood your journal’s policies, and we believe that neither the manuscript nor the study violates any of these. There are no conflicts of interest to declare.
Thank you for your consideration. I look forward to hearing from you.
2. Still the main problem is the small size of both groups which affects the statistical analyses. Perhaps it would be worthwhile to look for tests for small samples and not only non-parametric tests.
-Response: As you mentioned, the limitation of this study is the small sample size. To be honest, it was not easy to gather pregnant women in this study because of the specificity of the subjects and environmental problems such as Covid-19. For this reason, we referred to the title as a pilot study and this limitation in the discussion section.
3. Table 5 has a very low value, so it could be included in the appendix or omitted altogether. N is not given in the title, so perhaps these are two groups combined.
-Response: We agree with the reviewer’s advice and have therefore omitted table 5 in the manuscript.
4. PSQI test is misleading, as usually high quality means better outcome. Please remind in results section than high PSQI means worse sleep quality. And thus the imropvement in PE group is visible on the Figure 2.
-Response: PSQI is a questionnaire that evaluates the quality of the sleep index. As far as we know the closer to 0, the better the sleep quality, and the higher the number, the worse the sleep quality (R Rodriguez-Blanque et al., 2018). In our study, post-SQI scores had increased in the CON group compared with before. However, PSQI scores had significantly decreased in the PE group following HTPE in figure 2.
R Rodriguez-Blanque et al.,(2018). The influence of physical activity in water on sleep quality in pregnant women: A randomised trial. Women Birth. 31(1):e51-e58. doi: 10.1016/j.wombi.2017.06.018. Epub 2017 Jul 8.
5. It would also be better to annotate the p<0.001 value in Figure 2 instead of the ## marking, which is not readable at first glance.
-Response: As you suggested, we have revised all p values in Figure 2.
6. Since this is a preliminary study, it is worth adding further directions. Are there plans to implement such a program? What changes are worth making? In a study with a larger sample, a mixed model could also be proposed. In this study there was only pre-post comparison in each group separately.
-Response: As your suggestion, we referred to the limitation of our study especially the sample size at the end of the discussion. In addition, we have added further direction for the postpartum woman using our Pilates program. This is because the women who have given birth are very susceptible to stress which can induce postpartum depression.

This manuscript is a resubmission of an earlier submission. The following is a list of the peer review reports and author responses from that submission.
Round 1
Reviewer 1 Report
This study examined the effect of home-based tele-pilates exercise on low back pain and insomnia. The result is potentially beneficial. However, there are several methodological and ethical issues in this study. Below are my comments.
COI statement is missing.
The authors need to describe the clinical registration number. It is a standard practice to register the randomized control trial BEFORE a study begins.
The current sample size (each n = 7) is too small. How did the author determine this sample size? In the case of a small sample size, good results (i.e., significant) tend to be derived by chance. The authors need to describe its rationale clearly and consider conducting an additional experiment to increase n.
How did the authors assign the treatment randomly? In addition, was the blind condition established? The authors need to describe these aspects clearly.
The authors need to submit the CONSORT checklist with the manuscript.
The authors should describe the difference between two groups (Table 2-4) not in terms of z values but terms of Cohen’s d. The authors need to conduct a t-test and describe t-values with their df s(degree of freedom) substitute for z values.
Aside from background difference tests, the authors conducted many tests. The authors need to use a specific technique suitable for multiple comparisons, such as the Bonferroni method. In addition, the authors need to specify what the primary outcome is.
Author Response
The manuscript has been rechecked and the necessary changes have been made in accordance with the reviewers’ suggestions. The responses to all comments have been prepared and attached herewith.

Reviewer 2 Report
The paper is generally well done and well structured. The experimental design and statistic are both well performed.
The paper requires an English revision.
Title: the title is too long. I’d suggest to rethink the title in something like “The effect of a home-based tele-Pilates intervention on (Korean?) pregnant women”. I don’t think that the nationality is of interest in this case.
Line 107: is there an extra “C” here?
Line 108: please specify the Nation where this city belongs to.
Line 120-121: how did you chose this protocol? I mean the 50 minutes/day, twice a week, a total of 8 weeks, break duration etc…? Is there any previous research or community consensus?
Line 166-167: please specify the contraction regimen (typology of muscle strength test). It was an isometric muscle strength test I guess. Any reference for the isometric strength test protocol?
Line 168-169: how many times each participant did repeated the test (how many trials they performed)? It should be at least three times for the sake of intra-subject repeatability analysis. If just one time, please add this as limitation in the discussion section.
Add unit of measurement in Table 3 and 4. Moreover, please also modify the caption of Table 3 that solely says “Muscles of hip joint”! What is it? Strength values?
Discussion section: please discuss about the potential of using home-based, low cost and commercially available motion capture systems (such as exergames controllers (i.e., Microsoft Kinect) or wearable inertial sensors) to monitor the movement execution quantitatively for providing a real time biofeedback to the woman and for sending a report about the performed training to the medical doctor (tele-medicine). This scenario has been applied in a similar context related to sport and exercise sciences distance learning (please acknowledge the following paper https://doi.org/10.1007/978-3-030-31284-8_17 )
Author Response

(The authors gave the same response as above.)

Reviewer 3 Report
Hello.
Congratulations on your work. Virtual exercise programming certainly has become vitally necessary the last 2 years for the health and well-being of society. I have a few minor suggestions:
- Line 22-23- is "reduces body fat metabolism" correct? Should it be "increases?"
- Line 29-30: recommend using a different reference tailored more towards at risk groups
- Line 83- recommend qualifying this sentence- pregnant women are unable to participate in group exercise....
- Line 222-229- It is mentioned earlier in your paper but consider stating that you are measuring muscle strength in this section
- Line 259- remove the word "on"
- Line 336- consider rewording to "limited increased body fat," or simply decreased body fat
Best of luck!
Author Response

(The authors gave the same response as above.)

Round 2
Reviewer 1 Report
Thank you for your revision, but I am not fully satisfied with your responses. Below are my comments.
- I mean not the ethical committee’s number but the clinical trial registration number outside your institution. You can find registration examples at clinicaltrials.gov, UMIN-CTR, and so on.
- Considering that there is neither clinical registration nor sample size estimation in advance of the study, in addition to the small sample size of n = 7, most readers would not regard this study as full but preliminary. Thus, the authors need to add ”: A pilot study” or “:A preliminary study” to the end of the title.
- I mean not parametric or non-parametric methods but multiple comparisons (the repetition of the test). Such methods (e.g., Bonferroni) are usually involved in p-value adjustment and can be used with non-parametric and parametric tests. Please apply one of these adjustments to your statistical test.
- The non-parametric test usage is often accompanied by a median and quartile range report rather than Mean ± SD. In addition, please remove z-values.